# Unveiling the Stereoselectivity and Regioselectivity of the [3+2] Cycloaddition Reaction between N-methyl-C-4-methylphenyl-nitrone and 2-Propynamide from a MEDT Perspective

**DOI:** 10.3390/ijms24109102

**Published:** 2023-05-22

**Authors:** Sabir A. Mohammed Salih, Huda A. Basheer, Jesus Vicente de Julián-Ortiz, Haydar A. Mohammad-Salim

**Affiliations:** 1Faculty of Science, Department of Chemistry, University of Zakho, Duhok 42001, Iraqhayder.salim@uoz.edu.krd (H.A.M.-S.); 2Molecular Topology and Drug Design Research Unit, Department of Physical Chemistry, Pharmacy Faculty, University of Valencia, 46100 Valencia, Spain

**Keywords:** molecular electron density theory, nitrone, [3+2] cycloaddition reactions, electron localization function

## Abstract

[3+2] cycloaddition reactions play a crucial role in synthesizing complex organic molecules and have significant applications in drug discovery and materials science. In this study, the [3+2] cycloaddition (32CA) reactions of N-methyl-C-4-methyl phenyl-nitrone **1** and 2-propynamide **2**, which have not been extensively studied before, were investigated using molecular electron density theory (MEDT) at the B3LYP/6–311++G(d,p) level of theory. According to an electron localization function (ELF) study, N-methyl-C-4-methyl phenyl-nitrone **1** is a zwitterionic species with no pseudoradical or carbenoid centers. Conceptual density functional theory (CDFT) indices were used to predict the global electronic flux from the strong nucleophilic N-methyl-C-4-methyl phenylnitrone **1** to the electrophilic 2-propynamide **2** functions. The 32CA reactions proceeded through two pairs of stereo- and regioisomeric reaction pathways to generate four different products: **3**, **4**, **5**, and **6**. The reaction pathways were irreversible owing to their exothermic characters: −136.48, −130.08, −130.99, and −140.81 kJ mol^−1^, respectively. The enthalpy of the 32CA reaction leading to the formation of cycloadduct **6** was lower compared with the other path owing to a slight increase in its polar character, observed through the global electron density transfer (GEDT) during the transition states and along the reaction path. A bonding evolution theory (BET) analysis showed that these 32CA reactions proceed through the coupling of pseudoradical centers, and the formation of new C-C and C-O covalent bonds did not begin in the transition states.

## 1. Introduction

Nitrones are important three-atom components (TACs) commonly utilized in [3+2] cycloaddition (32CA) reactions, which have significant synthetic potential for the production of pharmacologically active isoxazolidines and isoxazolines that are stereochemically and regioselectively defined [1]. Over the last decade, numerous conceptual DFT (density functional theory) studies have been published on nitrone cycloadditions to demonstrate the productive interaction between experimental discoveries and DFT-based selectivity predictions [2,3,4,5,6,7,8,9,10]. From this perspective, communications published after 2003 have indicated the changes in electron density during the cycloaddition process and categorized TACs and ethyne derivatives according to the absolute electrophilicity scale [11]. Many mechanisms have been proposed for 32CA reactions since 2003. Polar mechanisms, such as two-stage, one-step processes, were suggested for 32CA reactions of nitrones in 2014 and 2017 [12,13,14,15]. In 2016, Domingo proposed the “molecular electron density theory” (MEDT), highlighting the impact of electron density changes in the reactivity of organic reactions [16]. MEDT studies have characterized the mechanism of 32CA reactions via sequential bonding changes along one-step reactions instead of the conventional concerted pericyclic pathway as proposed by Woodward and Hoffman in 1969. Parr functions are defined to explain regioselectivity in polar reactions, and many 32CA reactions have a non-polar character. In 2014, Tang et al. reported on theoretical DFT studies to investigate the mechanism of reactions between 1,3-dialkynes and ammonia derivatives [17,18,19,20]. In 2018, Jasiski suggested the competition of one-step and two-step processes in polar nitrone 32CA reactions to nitroethenes [21]. Recent theoretical publications from 2017 and 2018 have shown nitrone-32CA reactions to be non-concerted, one-step processes with asynchronous bond generation [17,18,22]. These 32CA reactions show different reactivity profiles; *pdr*-type 32CA reactions are associated with a lower energy barrier and take place easily, while *zw*-type 32CA reactions show a high energy barrier that needs to be overcome through electrophilic–nucleophilic interactions between the reactants [23].

The aim of this study is to explore the mechanism and energetics of the [3+2] cycloaddition reaction between N-methyl-C-4-methylphenyl-nitrone and 2-propynamide and to gain a better understanding of the electronic factors that control the regioselectivity and stereoselectivity of the reaction. The study is significant as it contributes to the development of more efficient and selective synthetic routes for the production of biologically active isoxazolidine and isoxazoline derivatives. The article is organized into five sections: (1) The determination of the electronic structures of the reagents N-methyl-C-4-methyl phenyl-nitrone **1** and 2-propynamide **2** by analyzing the electron localization function (ELF) [24,25]; (2) a conceptual density functional theory (CDFT) study of the reagents in their ground state to understand the electronic flow between them [3,26]; (3) an examination of the energy profile through the potential energy surfaces (PES) in the gas phase, THF, dichloroethane, benzene, and toluene along the possible regioisomeric pathways, followed to determine the effect of solvent polarity on the energy profile, and the determination of the global electron density transfer (GEDT) at the transition states (TSs) to forecast their polar character [6,27,28,29,30]; (4) an analysis of the intermolecular interactions at the TSs using a topological analysis of the ELF [31,32]; and (5) an investigation of the energetically acceptable reaction pathway via the bonding evolution theory (BET) [33,34,35].

## 2. Results and Discussion

### 2.1. Analysis of the ELF Topology of the Reactants N-methyl-C-4-methylphenyl-nitrone ***1*** and 2-Propynamide ***2***

This subsection provides an analysis of the electronic structure of the reactants N-methyl-C-4-methylphenyl-nitrone **1** and 2-propynamide **2** using the ELF and Natural Bond Orbital (NBO) methods. This analysis is crucial to achieving our research objective of understanding the factors that affect the reactivity of the reactants and the mechanism of the chemical reaction. The ELF, developed by Becke and Edgecombe, provides a precise mathematical representation of the electronic structure of a chemical system. Silvi and Savin later extended it to describe three localization attractors, namely, the core, bonding, and non-bonding attractors, to classify different electronic regions in a chemical system. The core basins, C(x), are the topological subdivisions of the ELF gradient field that surround the atomic nuclei; the monosynaptic valence basins, V(X), correspond to the non-bonding electron density of the lone pair or pseudoradical center of atom X; and the disynaptic basins, V(X,Y), correspond to the bonding region between X and Y [24,25]. The ELF topological analysis proposed by Domingo categorizes the three-atom components (TACs) involved in 32CA reactions based on ELF, namely, the pseudodiradical, pseudo(mono)radical, carbenoid, and zwitterionic TACs [16,36,37]. The domains of ELF localization and the most important valence basin populations of the B3LYP/6–311++G(d,p)-optimized reagents N-methyl-C-4-methylphenyl-nitrone **1** and 2-propynamide **2** are provided in Figure 1.

Table 1 shows the respective values of ELF valence basin populations in average number of electrons. The ELF analysis of N-methyl-C-4-methylphenyl-nitrone **1** reveals the existence of the V(O1) monosynaptic basin, which integrates 5.89 e and corresponds to the non-bonding electron density of the O1 oxygen. The disynaptic basins, V(C3, N2), contain a total population of 3.9 e associated with the C3-N2 double bond, and a V(N2, O1) disynaptic basin incorporates 1.34 e, related to the N2-O1 single bond. Notably, N-methyl-C-4-methylphenyl-nitrone **1** lacks a pseudoradical or carbine center and is classified as a zwitterionic TAC. The identification of the disynaptic basins associated with the C3-N2 and N2-O1 bonds can provide valuable insights into the potential reactive sites of the molecules in a 32CA reaction.

Likewise, the ELF analysis of 2-propynamide **2** shows the presence of disynaptic basins V(C4, C5) and V′(C4, C5), which combine to provide a total population of 5.28 e, associated with the C4-C5 triple bond (Table 1). This information can provide valuable insights into the potential reactivity of 2-propynamide **2** in a 32CA reaction.

The Lewis-like structures of N-methyl-C-4-methyl phenyl-nitrone **1** and 2-propynamide **2**, along with their NBO-derived charges, are shown in Figure 2. In N-methyl-C-4-methyl phenyl-nitrone **1**, the O1 oxygen has a negative charge of −0.54 e, while the C3 carbon has a positive charge of 0.03 e, indicating charge polarization in the framework. In 2-propynamide **2**, the C4 carbon has a negative charge of −0.16 e, and the C5 carbon is negatively charged by −0.11 e because of the conjugated carbonyl bond with the C-C triple bond moiety. These negative charges on O1 in N-methyl-C-4-methyl phenyl-nitrone **1** and C4 in 2-propynamide **2** make these atoms more susceptible to nucleophilic attack.

### 2.2. Analysis of the CDFT Indices

The reactivity indices of CDF can provide early insights into the direction of electronic flow between reagents [3,26]. In this study, the standard reactivity scales were determined using the B3LYP/6-31G(d) computational level; the present CDFT analysis was performed at this level. The resulting CDFT indices, Table 2, including the chemical hardness (η), electronic chemical potential (μ), electrophilicity (ω), and nucleophilicity (N) indices in eV, were then computed for two reagents: N-methyl-C-4-methylphenyl-nitrone **1** and 2-propynamide **2**, also at the B3LYP/6-31G(d) level of computation [3,13,38]. Our findings show that the electronic chemical potential, μ, of N-methyl-C-4-methylphenyl-nitrone **1** (μ = −3.71 eV) is higher than that of 2-propynamide **2** (μ = −4.55 eV), indicating the electronic flux from N-methyl-C-4-methylphenyl-nitrone **1** to 2-propynamide **2**. The electrophilicity (ω) index and the nucleophilicity (N) index of N-methyl-C-4-methylphenyl-nitrone **1** are 1.65 and 3.69 eV, respectively, classified as strongly electrophile and strongly nucleophile on their respective scales [13,38]. 2-propynamide **2**, with an electrophilicity index of ω = 1.64 eV, is classified as strongly electrophilic, and with nucleophilicity index (N = 1.79 eV), as weakly nucleophilic. Consequently, in *zw*-type 32CA reactions, 2-propynamide **2** will behave as an electrophile while N-methyl-C-4-methylphenyl-nitrone **1** will behave as a nucleophile, in conformity with the electronic chemical potentials, μ, of these species.

The chemical hardness (η) of N-methyl-C-4-methylphenyl-nitrone **1** (η = 4.16 eV) is lower than that of 2-propynamide **2** (η = 6.30 eV). Thus, the TAC N-methyl-C-4-methylphenyl-nitrone **1** is softer and more prone to electron density deformation compared with 2-propynamide **2**.

The chemical hardness index (η) and electrophilicity index (ω) are indicators of a molecule’s ability to either receive or donate electrons, respectively. Conversely, the nucleophilicity index (N) indicates a molecule’s electron-donating capacity. A higher value of N implies a stronger ability to donate electrons, while a lower value of ω indicates a stronger ability to accept electrons. In the studied reaction, the higher nucleophilicity of N-methyl-C-4-methylphenyl-nitrone **1** compared with 2-propynamide **2** suggests that the former is more likely to act as a nucleophile and donate electrons. This implies that the reaction may proceed through a nucleophilic addition mechanism, where the nucleophile attacks the electrophilic site of the other reactant.

Overall, the CDFT indices are a valuable source of information regarding the electronic properties and reactivity of these reactants. This information can be utilized to predict the reaction mechanism and kinetics with precision.

### 2.3. Analysis of the Potential Energy Surface along the Feasible Regioisomeric Pathways

The 32CA reaction of N-methyl-C-4-methylphenyl-nitrone **1** and 2-propynamide **2** can proceed through two regioisomeric routes, ortho and meta, depending on the attack of the nitrone oxygen on the C4 and C5 carbons of 2-propynamide **2** (Figure 1). By locating the stationary points along the PES of these two reaction pathways, it was possible to determine the position of the reagents N-methyl-C-4-methylphenyl-nitrone **1** and 2-propynamide **2**; the TSs (**TS1-en**, **TS1-ex**, **TS2-en**, and **TS2-ex**); and products **3**, **4**, **5**, and **6**. The energy profile study revealed several key findings:(i)The 32CA reaction of N-methyl-C-4-methylphenyl-nitrone **1** and 2-propynamide **2** shows negative reaction free energies from −45.96 to −79.36 kJ mol^−1^, including zero-point energy (ZPE) (see Appendix A), suggesting thermodynamic control and, hence, irreversibility.(ii)The ΔE of **TS2-en** is lower than that of **TS1-en** by 5.86, 6.07, 6.26, 7.14, and 6.9, in the gas phase, toluene, benzene, THF, and dichloromethane, respectively, suggesting exclusive **TS2-en** selectivity, in complete agreement with the experimental findings [38].(iii)**TS1-en** has an activation enthalpy of 80.52 kJ mol^−1^ in the gas phase, which increases to 90.12 kJ mol^−1^ in toluene, 87.50 kJ mol^−1^ in benzene, 94.29 kJ mol^−1^ in THF, and 94.82 kJ mol^−1^ in dichloroethane. This indicates an increase of 14.3 kJ mol^−1^ from the gas phase to dichloromethane, making the reaction energetically feasible in low-polarity solvents.

To determine the polar character, the GEDT was calculated at the TSs and is shown in Table 3. The located TSs exhibit minimal GEDT from 0.003 to 0.134 e, which is characteristic of forward electron density flux (FEDF). This suggests a partial polar character for the 32CA reaction [39].

The geometries of **TS1-en**, **TS1-ex**, **TS2-en**, and **TS2-ex** in the gas phase are illustrated in Figure 3. The distance between the interacting centers of C5 and O1 at **TS1-en** is greater than between C3 and C4 by 0.023 Å, and at **TS1-ex**, the difference is 0.013 Å. In **TS2-en**, the distance between C3 and C4 is larger than between C5 and O1 by 0.451 Å, whereas in **TS2-ex**, the difference is 0.524 Å. This suggests higher synchronicity in **TS2-en** and **TS2-ex** relative to **TS1-en** and **TS2-ex**.

When solvent effects such as toluene are considered, the distance between the interacting centers of C3 and C4 undergoes minor changes, ranging between 2.131 and 2.105 Å in **TS1-en**, 2.133 and 2.106 Å in **TS1-ex**, 2.335 and 2.457 Å in **TS2-en**, and 2.384 and 2.448 Å in **TS2-ex**, while the distance between the interacting centers of C5 and O1 ranges between 2.154 and 2.122 Å in **TS1-en**, 2.146 and 2.127 Å in **TS1-ex**, 1.884 and 1.789 Å in **TS2-en**, and 1.860 and 1.800 Å in **TS2-ex**.

### 2.4. Topological Analysis of the ELF at the TSs

The topological analysis of the ELF at the TSs allows for the evaluation of their electronic structure and the extent of the process of bond formation. Figure 4 illustrates how the ELF localization domains and basin attractor locations at the gas-phase TSs are related to the 32CA reaction.

The ortho-TSs, **TS1-en** and **TS1-ex**, show the presence of the V(O1) and V′(O1) monosynaptic basins, integrating a total population of 5.81e and 5.79 e, respectively. The meta-TSs, **TS2-en** and **TS2-ex**, show the presence of V(O1) and V′(O1), integrating 5.72 e and 5.72 e, respectively, and are connected to the density of non-bonding electrons at the O1 oxygen.

The ELF valence basin populations of these TSs are listed in Table 4. The ELF of **TS1-en**, **TS1-ex**, **TS2-en**, and **TS2-ex** shows the presence of V(C3,N2) disynaptic basins, integrating a total population of 2.44, 2.36, 2.69, and 2.83 e, respectively, which are associated with the C3–N2 bonding region. Additionally, the V(N2) monosynaptic basin integrates 1.39, 1.46, 1.45, and 1.31 e at **TS1-en**, **TS1-ex**, **TS2-en**, and **TS2-ex**, respectively, and is associated with the non-bonding electron density at the N2 nitrogen. Notably, the depopulation of the C3–N2 bonding region from 3.9 e at N-methyl-C-4-methylphenyl-nitrone 1 to 2.44, 2.36, 2.69, and 2.83 e at **TS1-en**, **TS1-ex**, **TS2-en**, and **TS2-ex**, respectively, is caused by a rupture in the C3-N2 double bond at the TSs.

The V(N2,O1) disynaptic basin is also depopulated from 1.34 e at N-methyl-C-4-methylphenyl-nitrone 1 to 1.21, 1.20, 1.15, and 1.18 e at **TS1-en**, **TS1-ex**, **TS2-en**, and **TS2-ex**, respectively. Consequently, the electron density of the V(N2) monosynaptic basin is mostly from the C3–N2 bonding region.

Moreover, the ELF of **TS1-en** and **TS1-ex** shows the presence of the V(C3) monosynaptic basin, integrating 0.37 and 0.33 e, respectively, which is connected to the formation of a pseudoradical center at C3 and is absent in **TS2-en** and **TS2-ex**. The ELF of **TS1-en**, **TS1-ex**, **TS2-en**, and **TS2-ex** also shows the presence of the V(C4,C5) and V′(C4,C5) disynaptic basins, integrating 4.49, 4.52, 4.50, and 4.38 e, respectively, associated with the C4–C5 bonding region. The depopulation of the C4–C5 bonding area from 5.28 e in 2-propynamide 2 to 4.49, 4.52, 4.50, and 4.38 e at the TSs establishes a pseudoradical center at C4, as shown by the existence of monosynaptic basin V(C4) integrating 0.59, 0.62, 0.73, and 0.76 e, respectively, at **TS1-en**, **TS1-ex**, **TS2-en**, and **TS2-ex**. At the ortho- and meta-TSs, the formation of a new single covalent bond has not yet begun; this was confirmed by computing the Wiberg bond index according to an NBO analysis [40] (see Appendix A).

Finally, the thermodynamic stability of cycloadducts 3 and 6 suggests regioselectivity, which is consistent with the experimental findings.

### 2.5. BET Study along the Favored Regiochemical Pathway

Krokoidis introduced the bonding evolution theory (BET) as a means of determining plausible chemical reaction mechanisms [33,41]. BET, in conjunction with the MEDT framework, was utilized to investigate the changes in electron density along the reaction pathway for the 32CA reaction of N-methyl-C-4-methyl phenyl-nitrone **1** and 2-propynamide **2**. This detailed examination of the bonding pattern along the ortho-regioisomeric pathways yielded seven ELF topological phases, each with its own typical IRC point (P1-I to **P6-I**), indicating the start of the phase. Figure 2 illustrates the phases and their starting points, P1-I through **P6-I**.

Phase I began at **P1-I** and displayed an ELF similar to that of the separate reagents (Figure 1). Phase II commenced at **P2-I**, with a bond length of d(O1–C5) = 2.656 Å and d(C3–C4) = 2.686 Å. The start of Phase III occurred at **P3-I**, with a bond length of d(O1–C5) = 2.154 Å and d(C3–C4) = 2.131 Å. During the 32CA reactions, the creation of a monosynaptic basin (V(N2)) was observed with an initial population of 1.39 e, which is associated with non-bonding electron density at the N2 nitrogen. It should be noted that the disynaptic basin (V(C3,N2)) was depopulated from 3.80 e at **P2-I** to 2.44 e at **P3-I**, resulting in the formation of new V(C3) and V(C4) monosynaptic basins with initial populations of 0.37 e and 0.59 e (as shown in Table 5). These basins were derived from the electron density within the C4–C5 bonding region, which experienced depopulation from 5.18 e at **P2-I** to 4.49 e at **P3-I** [42].

Phase IV begins at structure **P4-I**, where the distances of d(O1–C5) and d(C3–C4) are 1.892 Å and 1.814 Å, respectively. This phase is distinguished by the emergence of a new V(C5) monosynaptic basin with an initial population of 0.30 e (as documented in Table 5). Notably, the formation of the first C3–C4 single bond originates from the interplay of the pseudoradical centers at C3 and C4. It is also worth noting that, during this phase, the monosynaptic basins of V(C3) and V(C4) are absent, leading to a decline in the population of the C4–C5 bonding area from 4.49 e at **P3-I** to 4.02 e at **P4-I**.

Phase V commences at **P5-I**, where the distances of d(O1–C5) and d(C3–C4) measure 1.614 Å and 1.625 Å, respectively. The formation of the second O1–C5 single bond in this phase arises because of the merging of the pseudoradical center at C5 with a portion of the non-bonding electron density from the V(O) oxygen.

Ultimately, in cycloadduct **3**, the molecular geometry relaxes, resulting in a bond distance of d(O1–C5) at 1.458 Å and d(C3–C4) at 1.549 Å.

## 3. Computational Methods

The Berny analytical gradient optimization approach was employed to optimize stationary points along the potential energy surface of 32CA reactions at the B3LYP/6-311++G(d,p) level [43,44]. Previous studies have supported the use of the B3LYP functional as a reliable and accurate approach for several 32CA reactions [6,45,46,47]. Frequency calculations of the optimized TSs established the existence of one imaginary frequency, whereas the absence of an imaginary frequency for the local minimum was confirmed. To validate the minimal energy reaction pathway, IRC calculations were performed using the Gonzales–Schlegal integration technique. The established TSs connect the reactants and products. The solvent effects of toluene, benzene, THF, and dichloromethane were studied using the polarizable continuum model (PCM) and the self-consistent reaction field (SCRF) method [48,49,50,51]. The CDFT indices are computed using the equation discussed in References [2,3]. Natural population analysis (NPA) was used to construct the GEDT at the TSs of each reacting framework [28,52].
$$\mathrm{GEDT}=\sum q_{A}$$
where q stands for atomic charges, and the sum of charges in all atoms in the framework denotes the GEDT. A positive GEDT sign indicates global electronic flow from that framework to the other. ELF topological analyses were performed on the reagents [24,25].

The calculation of the global electrophilicity index (ω) was performed by using the formula ω = (µ^2^/2η) [53]. The chemical hardness (η) and electronic chemical potential (µ) were approximated in terms of one-electron energies of the Highest Occupied Molecular Orbital (ε_HOMO_) and Lowest Unoccupied Molecular Orbital (ε_LUMO_), as η ≈ **ε**_LUMO_ − **ε**_HOMO_ and µ ≈ (**ε**_HOMO_ + **ε**_LUMO_)/2, respectively [54,55]. The HOMO energies within the Kohn–Sham scheme were used to calculate the relative nucleophilicity N index [56,57]. This quantity may be defined as N = E_HOMO(Nu)_ − E_HOMO(TCE)_, and TCE (tetracyanoethylene) was chosen as the reference because it has the lowest HOMO energy [57]. TS and IRC isosurfaces were visualized using the program UCSF Chimera and computed using the Multiwfn program [58,59]. All computations were performed using the Gaussian 16 software [60].

## 4. Conclusions

The reaction mechanism of N-methyl-C-4-methyl phenyl-nitrone **1** and 2-propynamide **2** was studied at the B3LYP/6–311++G(d,p) level of theory using MEDT. The ELF topological investigation of the ground state structures classified N-methyl-C-4-methyl phenyl-nitrone **1** as a zwitterionic TAC, indicating its participation in *zw*-type 32CA reactions that require suitable electrophilic–nucleophilic interactions.

By comparing the electronic chemical potentials and nucleophilic strengths of N-methyl-C-4-methyl phenyl-nitrone **1** and 2-propanamide **2**, the global electron flow was predicted. This prediction was subsequently validated through GEDT calculations at the TSs, indicating that the 32CA reactions possessed negative free energy and were exergonic. Furthermore, the experimental regioselectivity was reflected in the higher thermodynamic stability of cycloadducts **3** and **6** compared with **4** and **5**.

BET analysis along the regioisomeric reaction channels allows for some significant mechanistic results. Along both reaction pathways, the C=N bonding region of N-methyl-C-4-methylphenylnitrone **1** and the triple bond bonding region of 2-propanamide **2** are depopulated in the first four phases to form pseudoradical centers at the C3, C4, and C5 carbons and lone pair electron density at the N2 nitrogen. ELF studies of the TSs further showed early TSs along the ortho- and meta-pathways before the formation of a new single covalent bond began. These findings provide a more profound comprehension of the mechanism of this type of reaction, which is valuable for designing more efficient and selective synthetic processes in organic chemistry.

## Data Availability

Not applicable.

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
