# Peer review of "Unveiling the Stereoselectivity and Regioselectivity of the [3+2] Cycloaddition Reaction between N-methyl-C-4-methylphenyl-nitrone and 2-Propynamide from a MEDT Perspective"

_ijms, 2023, doi:10.3390/ijms24109102_

Round 1

Reviewer 1 Report

Salih et al. utilized molecular electron density theory (MEDT) to investigate the [3+3] cycloaddition reaction of nitrone and propanamide in their article. Their comprehensive analysis of the intermediate and transition states using various approaches provides a valuable understanding of the fundamental principles of cycloaddition reactions. The authors used electron localization function (ELF), conceptual density functional theory (CDFT) parameters, and population analysis to examine the energetics and regioselectivity of the process in the gas phase and some dielectric media. The findings are well-presented and relevant, and the DFT method employed is sufficient for the calculations. However, the discussion section contains some inconsistencies and uncertainties that require revision.

Based on these observations, I recommend that the authors make significant revisions to the article before publishing it in IJMS. Addressing the concerns raised by the reviewer will ensure that the article meets the publication standards of the journal.

Major:

-       On page 3, line 96, the authors should explain the difference between V(C3, N2) and V′(C3, N2) in the text.

-       On page 6, line 159; the authors should explain why they chose a lower basis set for CDFT. Is there a specific reason to lower the level of theory in terms of computational cost? The reviewer suggests using some diffuse function and polarizability to get a better electronic property such as HOMO and LUMO position in the energy landscape.

-       On page 6, line 168-169; the statement ′′Additionally, both N-methyl-C-4-methylphenyl-nitrone 1 (ω = 1.65 168 eV) and 2-propynamide 2 (ω = 1.64 eV) are classified as strong electrophiles.” I am a little confused to know that compound 1 is a strong nucleophile as well as a strong electrophile. The authors should rephrase or explain it.

-       On page 6, lines 174-174; the authors should explain how chemical hardness index (η) is related to the ability to donate/receive electrons.

-       Page 6, lines 195-197; the statement “The 32CA reaction of N-methyl-C-4-methylphenyl-nitrone 1 and 2-propynamide 2 has negative reaction free energies ranging from -45.96 to -79.36 kJ mol-1, indicating kinetic control and irreversibility. “    is incorrect. Negative reaction free energy indicates thermodynamic control/favorability, whereas kinetic control comes from the reaction barrier. The authors should rephrase the sentences accordingly.

-       Page 6, lines 198-202; the statement “The ΔE of TS1-en is higher than that of (TS1-ex by 2.54, 1.72, 2.01, 2.38, and 2.11, 198 TS2-en by 5.86, 6.07, 6.26, 7.14, and 6.9, and TS2-ex by 1.69, 2.18, 2.43, 4.25, and 199 3.89 kJ mol-1) in the gas phase, toluene, benzene, THF, and dichloromethane, respectively. This suggests exclusive TS1-en selectivity, which aligns with experimental findings” is confusing. According to Table 3, TS2-en has the lowest activation barrier in terms of both ΔE and ΔH, and hence it should be the favorable pathway. The authors should clarify this point. Additionally, the reviewer suggests using a superscript (‡) to denote activation energies, as this is commonly used in the literature.

-       The reviewer recommends the authors use at least the enthalpy (if not free energies) to compare results to incorporate zero-point correction (if not entropy correction).

-       Scheme 1 needs correction. In 2-propynamide (2), C5 is substituted and C4 is the terminal carbon. However, in TS2s, C5 is unsubstituted. The authors should correct the labels for better understanding.

-       Page 8, lines 262-263; “At the ortho and meta TSs, the formation of a new single covalent bond has not yet begun.” It would be helpful if the authors could confirm whether any significant bond order has developed in the ortho and meta TSs by checking the bond index from the NBO calculation.

-       In order to provide more detailed information, the authors may consider including a supporting file that contains the optimized coordinates and electronic energies of all the intermediates and transition states. This could be useful for readers who wish to reproduce the calculations or conduct further analysis.

Minor:

-       On page 2 lines 45-48; “MEDT studies have characterized the mechanism of 32CA reactions by consecutive bonding changes along a one-step reaction, rather than the conventional concerted pericyclic pathway suggested by Woodward and Hoffman in 1969, Parr operations have been developed to explain regioselectivity in polar reactions.” Rewrite this to better understand.

-       On page 2 lines 48-50; “Although many 32CA reactions are non-polar, Tang et al. used theoretical DFT to study the mechanism of reactions between 1,3-alkynes and ammonia derivatives in 2014.” This needs to be rephrased as I do not understand why the author used “non-polar” to begin the sentence.

The English used is generally clear and comprehensible, although certain parts may require some rephrasing. My feedback to the authors includes specific suggestions for improvement. Therefore, I recommend that the authors carefully review and revise their writing.

Author Response

We wish to express our appreciation to the Reviewers for their insightful comments, which have helped us significantly to improve our manuscript. According to the suggestions, we have thoroughly revised our manuscript and its final version is enclosed. Point-by-point responses to the comments are listed below.

Reviewer 1

Comment 1:

 On page 3, line 96, the authors should explain the difference between V(C3, N2) and V′(C3, N2) in the text.

Response to Comment 1:                

The sentence has modified. There is only one disynaptic basin with total population of 3.9 indication the double bond region between N2-C3.

Comment 2:  

 On page 6, line 159; the authors should explain why they chose a lower basis set for CDFT. Is there a specific reason to lower the level of theory in terms of computational cost? The reviewer suggests using some diffuse function and polarizability to get a better electronic property such as HOMO and LUMO position in the energy landscape.

Response to Comment 2:                

The electrophilicity and nucleophilicity scales were performed at B3LYP/6-31g(d) level of theory by Domingo in 2002 and 2011, see References 13 and 39.

Comment 3:

   On page 6, line 168-169; the statement ′′Additionally, both N-methyl-C-4-methylphenyl-nitrone 1 (ω = 1.65 168 eV) and 2-propynamide 2 (ω = 1.64 eV) are classified as strong electrophiles.” I am a little confused to know that compound 1 is a strong nucleophile as well as a strong electrophile. The authors should rephrase or explain it.

Response to Comment 3:                

The sentence has modified and highlighted in yellow. According to the electrophilicity and nucleophilicity scales performed by Domingo, see Reference 13 and 39, this nitrone has a dual characters. 

Comment 4:

On page 6, lines 174-174; the authors should explain how chemical hardness index (η) is related to the ability to donate/receive electrons.

Response to Comment 4:

The sentence is added and highlighted in yellow explaining the chemical hardness.

Comment 5:

Page 6, lines 195-197; the statement “The 32CA reaction of N-methyl-C-4-methylphenyl-nitrone 1 and 2-propynamide 2 has negative reaction free energies ranging from -45.96 to -79.36 kJ mol-1, indicating kinetic control and irreversibility. “ is incorrect. Negative reaction free energy indicates thermodynamic control/favorability, whereas kinetic control comes from the reaction barrier. The authors should rephrase the sentences accordingly.

Response to Comment 5:

The statement has modified and highlighted in yellow. Thanks.

Comment 6:

Page 6, lines 198-202; the statement “The ΔE of TS1-en is higher than that of (TS1-ex by 2.54, 1.72, 2.01, 2.38, and 2.11, 198 TS2-en by 5.86, 6.07, 6.26, 7.14, and 6.9, and TS2-ex by 1.69, 2.18, 2.43, 4.25, and 199 3.89 kJ mol-1) in the gas phase, toluene, benzene, THF, and dichloromethane, respectively. This suggests exclusive TS1-en selectivity, which aligns with experimental findings” is confusing. According to Table 3, TS2-en has the lowest activation barrier in terms of both ΔE and ΔH, and hence it should be the favorable pathway. The authors should clarify this point. Additionally, the reviewer suggests using a superscript (‡) to denote activation energies, as this is commonly used in the literature.

Response to Comment 6:

The note is well-taken. The sentence is modified and highlighted in yellow. Thanks.

Comment 7:

The reviewer recommends the authors use at least the enthalpy (if not free energies) to compare results to incorporate zero-point correction (if not entropy correction).

Response to Comment 7:

The comment is well-taken. A sentence is added regarding the ZPE and all calculations, in a.u., were added to the see Supplementary Materials file.

Comment 8:

Scheme 1 needs correction. In 2-propynamide (2), C5 is substituted and C4 is the terminal carbon. However, in TS2s, C5 is unsubstituted. The authors should correct the labels for better understanding.

Response to Comment 8:

The note is well-taken. The Scheme has modified.

Comment 9:

Page 8, lines 262-263; “At the ortho and metaTSs, the formation of a new single covalent bond has not yet begun.” It would be helpful if the authors could confirm whether any significant bond order has developed in the ortho and metaTSs by checking the bond index from the NBO calculation.

Response to Comment 9:

The Wiberg bond index were calculated for all TSs to confirm whether any significant bond order has developed in the TSs. All calculation were listed in the Supplementary Materials file.

Comment 10:

In order to provide more detailed information, the authors may consider including a supporting file that contains the optimized coordinates and electronic energies of all the intermediates and transition states. This could be useful for readers who wish to reproduce the calculations or conduct further analysis.

Response to Comment 10:

A Supplementary Materials file is prepared and will be attached during submission.

Comment 11:

Minor:

-       On page 2 lines 45-48; “MEDT studies have characterized the mechanism of 32CA reactions by consecutive bonding changes along a one-step reaction, rather than the conventional concerted pericyclic pathway suggested by Woodward and Hoffman in 1969, Parr operations have been developed to explain regioselectivity in polar reactions.” Rewrite this to better understand.

Response to Comment 11:

The sentence has modified and highlighted in yellow.

Comment 12:

  On page 2 lines 48-50; “Although many 32CA reactions are non-polar, Tang et al. used theoretical DFT to study the mechanism of reactions between 1,3-alkynes and ammonia derivatives in 2014.” This needs to be rephrased as I do not understand why the author used “non-polar” to begin the sentence.

Response to Comment 12:

The sentence has modified and highlighted in yellow.

Reviewer 2 Report

The manuscript is devoted to the theoretical study of the detailed mechanism of [3+2] cycloaddition (32CA) reactions of N-methyl-C-4-methyl phenyl-nitrone 1 and 2-propynamide 2. A number of 32CA reactions of N-methyl-C-4-methyl phenyl-nitrone 1 are studied during many years due to the significant synthetic potential for the production of pharmacologically active isoxazolidines and isoxazolines. The topicality of the research deals with the importance to regulate stereo- and regioselectivity of the reaction. A number of modern quantum-chemical approaches was applied such as conceptual density functional theory (CEDT), examination of the energy profile through the potential energy surfaces (PES), molecular descriptors, etc.

The main results are:

·       The ELF topological investigation of the ground state structures classified N-methyl-C-4-methyl phenyl- nitrone 1 as a zwitter-ionic molecule, indicating its participation in zw-type 32CA reactions that require electrophilic-nucleophilic interactions.

·       In reaction of N-methyl-C-4-methyl phenyl- nitrone 1 with 2-propanamide 2, the global electron flow was predicted.

·       BET analysis along the regioisomeric reaction channels allows to analyze mechanistic details of 32CA reactions.

The title of the manuscript, as well as its bibliography corresponds to its content.

Only minor revisions should be done, in my opinion. For example, in the introduction, the importance of the study was explained at least twice:

Nitrones are important three-atom-components (TACs) commonly utilized in [3+2] 32

cycloaddition (32CA) reactions, which have significant synthetic potential for the produc-33

tion of pharmacologically active isoxazolidines and isoxazolines that are stereochemically 34

and regioselectively defined [1].

In organic chemistry, C-aryl-N-methyl nitrones 1 are used 54

as starting materials to produce various new isoxazolidine heterocycles. C-aryl-N-alkyl 55

 nitrones have been used to create new bioactive chemicals [23].

It is better to join them in one paragraph.

Author Response

We wish to express our appreciation to the Reviewers for their insightful comments, which have helped us significantly to improve our manuscript. According to the suggestions, we have thoroughly revised our manuscript and its final version is enclosed. Point-by-point responses to the comments are listed below.

Reviewer 2

Comment 1:

Nitrones are important three-atom-components (TACs) commonly utilized in [3+2] 32 cycloaddition (32CA) reactions, which have significant synthetic potential for the produc-33 tion of pharmacologically active isoxazolidines and isoxazolines that are stereochemically 34 and regioselectively defined [1].

In organic chemistry, C-aryl-N-methyl nitrones 1 are used 54

as starting materials to produce various new isoxazolidine heterocycles. C-aryl-N-alkyl 55 nitrones have been used to create new bioactive chemicals [23].

Response to Comment 1:

The note is well-taken. The second sentence has been replaced and highlighted in yellow.

Round 2

Reviewer 1 Report

The concerns I raised have been addressed by the author, and the necessary changes have been incorporated into the revised version of the manuscript. Based on this, I believe the manuscript is now ready for publication in the  IJMS.